

# Fungi present distinguishable isotopic signals in their lipids when grown on glycolytic versus tricarboxylic acid cycle intermediates

Stanislav Jabinski[1,2] Vítězslav Kučera[3,4], Marek Kopáček[1,5], Jan Jansa[4], Travis B. Meador[1,2,5*]

[1]University of South Bohemia, Faculty of Science, Department of Ecosystem Biology, Branišovská 1760, 370 05 České Budějovice, Czechia

[2]Institute of Soil Biology and Biochemistry, Biology Centre CAS, Na Sádkách 7, 370 05 České Budějovice, Czechia

[3]Charles University, Faculty of Sciences, Albertov 6, 128 00 Praha, Czechia

[4]Institute of Microbiology CAS, Vídeňská 1083, 142 20 Praha, Czechia

[5]Institute of Hydrobiology, Biology Centre CAS, Na Sádkách 7, 370 05 České Budějovice, Czechia

*Correspondence to*: Travis B. Meador (travis.meador@bc.cas.cz)

**Abstract.** Microbial activity in soils controls both the size and turnover rates of large carbon (C) inventories stored in the subsurface, having important consequences for partitioning of C between terrestrial and atmospheric reservoirs as well as recycling of mineral nutrients such as nitrogen or phosphorus (often bound to the C) that support plant growth. Fungi are major decomposers of soil organic matter (SOM); however, uncertainty about the identity of predominant C substrates that fuel their respiration confound models of fungal production and SOM turnover. To further define the signals of microbial heterotrophic activity, we applied a dual hydrogen (H) and C stable isotope probing (SIP) approach on pure fungal cultures representing the phyla Ascomycota, Basidiomycota, and Zygomycota growing on monomeric (glucose, succinate) or complex substrates (tannic acid, β-cyclodextrin). Our findings demonstrate that the investigated species incorporated only minor amounts of inorganic C (provided as bicarbonate) into their membrane lipids, amounting to < 3% of lipid-C, with no consistent patterns observed between species or growth substrates. The net incorporation of water-derived H (i.e., $a_W$) into lipids also did not differ significantly between incubations with monomeric versus complex substrates; however, growth on succinate solicited significantly higher $a_W$ values than glucose or β-cyclodextrin. This finding suggests that $^2$H-SIP assays have the potential to distinguish between microbial communities supported predominantly by substrates that are catabolized by the tricarboxylic acid cycle versus glycolytic pathway. Furthermore, the average $a_W$ value of heterotrophic fungal incubations [0.69 ± 0.03 (SEM)] is consistent with that observed for bacterial heterotrophs, and may be applied for upscaling lipid-based estimates of fungal production in environmental assays.



**Short Summary**
Microbial production is a key parameter in estimations of organic matter cycling in environmental systems, and fungi play a
major role as decomposers. In order to investigate fungal production and turnover times in soils, we incubated fungal pure
cultures with isotopically labelled water and bicarbonate to investigate growth signals encoded into lipid biomarkers, which
can be applied to improve flux estimates in environmental studies.

**1 Introduction**
Soil organic matter (SOM) is the major reservoir of carbon ($1580 \times 10^{15}$ g C) in the biosphere, and active microbial populations
act to redistribute this C to other reactive reservoirs, such as the atmosphere (Carson et al., 2001; Grinhut et al., 2007). Major
uncertainties in modeling C and climate dynamics stem from insufficient knowledge on the controls of SOM degradation and
transformation (Ciais et al., 2014; Lindahl and Tunlid 2015). Saprotrophic soil fungi are one of the major decomposers in soils,
are known to degrade naturally occurring complex molecules such as lignin (Kirk & Farrell, 1987; Fioretto et al., 2005;
Baldrian et al., 2011), cellulose (Šnajdr et al., 2011) and humic substances (Grinhut et al., 2007), but are also reported to
compete for accessible plant photosynthate excreted by roots (De Boer et al., 2005; Högberg et al., 2001; Smith & Read, 2008).
Despite the unique and important fungal niche in biogeochemical cycles, their contributions to SOM cycling remains poorly
constrained (Frey 2019; Grinhut et al., 2007). Furthermore, heterotrophic organisms feeding on organic substrates to gain
energy and build biomass are also known to fix a variable amount of inorganic C (IC), in order to replenish intermediates in
the tricarboxylic acid (TCA) cycle (Kornberg 1965). It has been suggested that 2 - 8% of the biomass C in heterotrophs
originates from IC incorporated through anaplerotic carboxylation reactions (Romanenko 1964; Roslev et al., 2004; Braun et
al., 2021). Awareness of these processes has existed for decades (Kornberg 1965; Sorotkin 1966). Yet, the relevance and
metabolic controls on heterotrophic IC fixation remains poorly understood, partly due to the lack of reliable estimates for most
organisms and habitats (Braun et al., 2021).
Advanced analytical techniques now allow microbial taxa to be linked to specific processes in environmental studies by
measuring incorporation of stable isotopes into biomarkers (Boschker et al., 1998; Dumont and Murrell, 2005; Kreuzer-Martin,
2007), such as fungal and bacterial membrane lipid fatty acids (Treonis et al., 2004; Willers et al., 2015) or other biomarkers
(Boschker and Middelbourg, 2002). Previous studies have demonstrated that variability in the $^{2/1}$H composition of microbial
lipids is redundant with that of environmental water (Hoefs, 2018; Kopf et al., 2015), and stable isotope probing (SIP) assays
applying enrichments in $^{2}H_2O$ have proven to be a useful tracer of microbial activity in a diverse range of environments
(Canarini et al., 2024; Caro et al., 2023; Fischer et al., 2013; Kellermann et al., 2012; Wegener et al., 2016; Warren 2022; Wu
et al., 2018). Large H-isotope fractionations, yielding $\delta^2$H values between −400‰ and +200‰, have been observed during
biosynthetic incorporation of water hydrogen (water-H) into individual compounds within a single cell or total biomass, which
can be indicative of the underlying metabolic processes (Osborn et al., 2011; Sachse et al., 2012; Zhang et al., 2009). To fully
exploit the potential of SIP experiments, a dual-SIP approach was developed to track total microbial production by adding



heavy water ($^2H_2O$) together with $^{13}C$-labeled IC, enabling simultaneous estimates of total and autotrophic metabolism,
respectively (Wegener et al., 2012; Wu et al., 2020). Recently, Jabinski et al. (2024) validated an innovation of the dual-SIP
assay by using rapid pyrolysis of fungal biomass to determine the stable C and H isotopic composition of fungal lipids, and
demonstrated that water-H and IC assimilation signatures could successfully distinguish between fungal ecotypes growing on
glucose or glutamic acid as the C source. The aim of the current study was to further assess the controls on water-H and IC
incorporation into lipids and expand our knowledge for interpreting environmental signals by applying the dual-SIP assay on
a broader range of pure fungal cultures and growth substrates, including labile monomers versus more complex, high molecular
weight molecules. We hypothesized that (i) the incorporation of IC and water-H into the fungal fatty acid biomarker $C_{18:2}$ will
be similar for fungal species growing on the same substrate, and (ii) that IC and water-H incorporation will distinguish between
growth on labile versus more complex C substrates.

## 2 Methods

### 2.1 Cultivation & Harvesting

Fungal pure cultures of two Basidiomycota [*Paxillus involutus* (PI, strain SB-22); *Phanerodontia chrysosporium* (PC, strain
CCM8074)], two Zygomycota [*Mortierella* sp. (MO, strain RK-38); *Umbelopsis* sp. (UM, strain RK-43)] and two Ascomycota
[*Penicillium janczewskii* (PJ, strain BCCO20_0265); *Paecilomyces lilacinus* (PL, strain DP-23)] were incubated in 500 mL
Schott bottles at 25 °C in the dark. Liquid mineral media (50 mL) was adapted after Bukovská et al. (2018) with the vitamins
left out, and was inoculated with approximately $10^6$ spores, or, for Basidiomycota, a hyphal block < 0.5 cm³ recovered from a
previous culture using the same cultivation medium solidified with agar (1.5%).
The growth medium contained per liter: 4 g organic C in various forms ($C_6H_{12}O_6$ glucose; $C_4H_6O_4$ succinic acid; $C_{42}H_{70}O_{35}$ β-
Cyclodextrin or $C_{76}H_{52}O_{46}$ tannic acid), 0.01 g $FeSO_4 * 7H_2O$, 2 g $KH_2PO_4$, 0.5 g $MgSO_4 * 7H_2O$, 0.1 g NaCl, 0.1 g CaCl, 2.5
g $(NH_4)_2SO_4$, 0.45 g $NaHCO_3$ and 1 mL of a mixed solution (per liter: 0.5 g $H_3BO_3$, 0.04 g $CuSO_4 * 5H_2O$, 0.1 g KI, 0.4 g
$MnSO_4 * 5H_2O$, 0.2 g $NaMoO_4 * 2H_2O$, 0.4 g $ZnSO_4 * 7H_2O$). The pH of the medium was adjusted to 4.5 before inoculation.
Dual-SIP experiments were performed using $^{13}C$-bicarbonate ($^{13}C$-DIC, $NaH^{13}CO_3$) and deuterated water ($D_2O$). Each fungal
strain was grown in triplicate with non-labeled substrates (Treatment I), with $δ^2H$ of the medium water adjusted to 100‰ and
$AT^{13}C$ = 10% of $^{13}C$-DIC (Treatment II), 200‰ $δ^2H$ and 10% $^{13}C$-DIC (Treatment III), and 400‰ $δ^2H$ and 10% $^{13}C$-DIC
(Treatment IV). The concentration of the bicarbonate in the cultivation medium was 0.1 g $L^{-1}$. The Schott bottles were closed
with a rubber stopper in order to prevent the labeled $^{13}C$-DIC from outgassing, and ample headspace was provided to maintain
oxic conditions throughout the growth experiment. Fungal growth was monitored via the accumulation of $CO_2$ in the
headspace, and we aimed to harvest when $CO_2$ levels reached 10%; however, without preliminary knowledge of the fungal
growth dynamics, some cultivations exceeded this level more quickly than they could be harvested.



To harvest the fungal biomass, mycelia were separated from the growth medium via vacuum filtration through 5 μm Isopore
polycarbonate filters (47 mm diam, Merck catalogue number TMTP04700) and the cultivation medium was collected into a
sterile 50 mL tube. Thereafter, the mycelium was washed with ample MilliQ water, transferred to pre-weighed, sterile 50 mL
tubes; the fresh weight of the biomass was recorded, and the samples were frozen at -80 °C until lyophilization. A subsample
of the cultivation medium was also frozen at -80 °C and the rest was used to determine pH post-incubation. After lyophilization,
the dry weight of each sample was determined and stored at -20 °C until further analysis.

**2.2 Measurements**
**2.2.1 Headspace $CO_2$ concentration and isotope composition**
Samples of headspace (0.3 mL) were collected weekly from each bottle into helium flushed 12 mL exetainer vials (Exetainer,
Labco Limited, UK) and analyzed for their $CO_2$ concentration and $^{13}C/^{12}C$ isotopic ratio using GasBench II equipped with a
single cryo-trap connected to Delta V Advantage isotopic ratio mass spectrometer (IRMS) via Conflo IV (Thermo Scientific,
Bremen, Germany). Ambient air (with its $CO_2$ concentration measured using LiCor 850 gas analyzer previously) was used as
a standard for $CO_2$ concentration measurements, whereas a laboratory cylinder with 0.1% $CO_2$ in helium was used as a standard
for isotopic composition ($\delta^{13}C$ = -2.86 ‰). The analytical precision was below 1‰. Data were analyzed and exported using
Isodat 3.0 software.
**2.2.2 Medium water ($^2H_2O$)**
Liquid samples were transferred into 1.5 ml glass vials (32 x 11.6 mm, Fischer Scientific) and then measured using Triple
Liquid Water Isotope Analyzer (Los Gatos Research), which is based on the principle of high-resolution laser absorption
spectroscopy. Samples were dispensed into the instrument using an autosampler (PAL3 LSI, ABB company) and a 1.2 μL
syringe (Hamilton). Samples were measured and evaluated against prepared laboratory standards of known isotopic
composition. The isotopic ratios of these laboratory standards were verified by measuring against international standards
(VSMOW2, SLAP2) made by the IAEA. For quality control purposes, the measurements of the samples were also interspersed
with periodic measurements of the prepared verification samples with known isotopic composition. The final isotopic
composition ($\delta^2H$) was determined using LIMS software. Analytical precision of $\delta^2H$ was <1.5‰.
Water sampled from incubations with tannic acid could not be measured using the laser, as described above, due to its high
organic carbon content, and was rather measured via a GasBench II device (Thermo Scientific, Bremen, Germany; Application
Note: 30049). Medium water samples (200 μL) were added with a platinum catalyst to a 12 mL exetainer vials (Exetainer,
Labco Limited, UK). The headspace was flushed with 1% $H_2$ in He at approximately 100 mL min$^{-1}$ for 6 min. After an
equilibration time of over 40 min, the samples were measured by purging the exetainer using a double-holed needle with
helium into a 250 μL sample loop. The sample was then injected and separated via a Carboxen PLOT 1010 (0.53 mm ID;



Supelco, Bellefonte, USA) held at 90 °C with a flow rate of ~2.2 mL min⁻¹, and then introduced into the MAT253 Plus IRMS
via a Conflo IV interface. Each sample was injected three times during one analysis. The isotopic composition was determined
using Isodat 3.0 software against the corresponding $H_2$ working gas (-239‰ for $\delta^2H$) and the values were corrected and
normalized using international standards VSMOW2 (0‰ for $\delta^2H$), SLAP2 (-427.5‰ for $\delta^2H$), USGS53 (+40.2‰ for $\delta^2H$) and
GFLES-2 (159.9‰ for $\delta^2H$). The analytical precision was around 1‰.

**2.2.3 Carbon ($\delta^{13}C$) substrate analysis**
Substrates (~100 µg) were weighed into tin capsules (8 × 5 mm, Sercon, Crewe, UK) and placed in a helium-flushed carousel
autosampler, then introduced to an Elemental Analyzer IsoLink device (EA IsoLink CNSOH, Thermo Scientific, Bremen,
Germany) equipped with a CHN/NC/N EA combustion/reduction reactor (Sercon, Crewe, UK) heated to 1020 °C. A pulse of
oxygen was introduced to the reactor simultaneously with the sample. The sample gases were quantified via a thermal
conductivity detector (TCD) and then introduced to a MAT 253 Plus isotope ratio mass spectrometer (IRMS; Thermo
Scientific; Bremen, Germany) via the open split of a Conflo IV interface, with helium as the carrier gas. The isotopic
composition was determined using Isodat 3.0 software against the corresponding $CO_2$ working gas (-4.191‰ for $\delta^{13}C$), and
the values were corrected for linearity and normalized to the VPDB scale using international reference material IAEA-600 (-
27.771‰ for $\delta^{13}C$). The analytical precision was <0.04‰.
**2.2.3 Pyrolysis GC for lipid analysis**
The pyrolysis unit Shimadzu 3030D (Shimadzu, Kyoto, Japan/ Frontier Laboratories, Fukushima, Japan) was installed on top
of the GC Trace1310 gas chromatograph SSL injector (Thermo Scientific, Bremen, Germany) and the GC was equipped with
an SLB-IL60 column (non-bonded; 1,12-Di(tripropylphosphonium)dodecane bis(trifluoromethanesulfonyl)imide phase, 30 m,
0.25 mm ID, 0.20 µm df, Supelco, Bellefonte, USA). The furnace temperature was 650 °C and the interface temperature was
370 °C. The injector temperature was 360 °C and the GC oven was held at 80 °C for 1 min then ramped to 175 °C at 15 °C
min⁻¹, then ramped to 195 °C at 2 °C min⁻¹, then ramped to 300 °C at 10 °C min⁻¹, and finally held at 300 °C for 7 min. Helium
was used as the carrier gas, with a constant flow of 1.5 mL min⁻¹, a split ratio of 40, and a split flow of 26.7 mL min⁻¹. The
column flow was split via a multichannel device to acquire MS and isotopic data simultaneously from one injection. The
GCMS (ISQ QD; Thermo Scientific, Bremen, Germany) ion source was set to electron impact ionization mode (EI) at 70 eV
and a scan range of m/z 50 – 500 with a scan time of 0.2 sec⁻¹. Scanning started after 8 min to avoid the solvent peak in the
MS. The transfer line temperature was set to 300 °C and the ion source was set to 250 °C.
The samples (lyophilized biomass, 0.1 – 1.3 mg) were weighed into an ultra-clean stainless steel Eco-Cup LF (Frontier
Laboratories, Fukushima, Japan), which were burned with a torch before usage to remove contaminants. Immediately prior to
sample injection, 30 µL of trimethylsulfonium hydroxide (TMSH) was added on the sample to increase the volatization of the



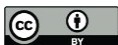

fatty acids and improve measurement sensitivity. Identification of fatty acid methyl esters (FAMEs) was performed using
fragmentation patterns and the NIST 14 library.
Stable carbon and hydrogen isotope compositions of FAMEs were determined by splitting the flow from the GC column to a
GC-IsoLink II reactor, coupled to a MAT253 Plus IRMS via a Conflo IV interface. Values are expressed in standard delta
notation ($\delta^{13}$C and $\delta^2$H). MS information was simultaneously acquired by use of the multi-channel device described above.
For conversion of FAMEs and ergosterol to $CO_2$, the combustion reactor (nickel oxide tube with CuO, NiO, and Pt wires) was
set to 1000 °C. For conversion of FAMEs and ergosterol to $H_2$, the pyrolysis reactor (aluminum tube) was set to 1420 °C. The
FAMEs were identified by their retention times and fragmentation patterns. The isotopic composition was determined using
Isodat 3.0 software against the corresponding $CO_2$ or $H_2$ working gas (-4.191‰ for $\delta^{13}$C, -239.5‰ for $\delta^2$H). Isotope corrections
for instrument drifts, linearity, and normalization to the VPDB or VSMOW scales were performed according to the response
of USGS70 (-30.53‰ for $\delta^{13}$C, -183.9‰ for $\delta^2$H) and USGS72 (-1.54‰ for $\delta^{13}$C, 348.3‰ for $\delta^2$H) reference standards. The
analytical precision was < 0.5‰ and < 10‰ for $\delta^{13}$C and $\delta^2$H, respectively.

**2.3 Calculations**
Carbon use efficiency (CUE) for the growth experiments was calculated by normalizing the amount of C in biomass by that
plus C that accumulated as $CO_2$ [CUE = biomass-C / ($CO_2$-C + biomass-C)]. The $\delta^{13}$C values of fungal biomarker $C_{18:2}$ was
determined as standard delta values (‰). The IC incorporation into the biomarker (%IC) was calculated based on the following
equation:
$$\%IC_{(assimilation)} = \frac{^{13}F_{lipid} - {}^{13}F_{lipid,control}}{^{13}F_{DIC(medium)} - {}^{13}F_{substrate}} \times 100 \qquad \textbf{(Eq. 1)}$$
**Equation 1: Inorganic carbon (%IC) assimilation was calculated as the difference in the $^{13}$C atom fraction of the lipids harvested**
**at the end of the labeling experiment ($^{13}F_{lipid}$) compared to the lipids harvested at the end of the natural abundance treatment**
**($^{13}F_{lipid,control}$), relative to the difference between the mixing-weighted average $^{13}$C atom fraction of dissolved inorganic C ($^{13}F_{DIC}$, cf.**
**Text S1) and the $^{13}$F of the substrate. F was calculated as $^{13}F = (R^{13/12})/(R^{13/12} + 1)$, where R is re-calculated from the $\delta^{13}$C ratios**
**reported by Isodat Software following measurement by IRMS ($\delta^{13}$C = ([$R^{13/12}$]$_{sample}$/[$R^{13/12}$]$_{ref}$ - 1) * 1000 (modified after Boschker**
**& Middelburg 2002; Wegener et al., 2012).**
The water H assimilation efficiency ($a_W$) for fungal biomarker $C_{18:2}$ was approximated from the regressions of the hydrogen
isotope composition of individual fatty acids $^2F_{FA}$ and that of medium water ($^2F_{water}$), according to Kopf et al. (2015). Briefly,
the hydrogen isotopic compositions of microbial fatty acids produced by an organism generally follow the isotopic composition
of environmental water, and are related to the mole fraction of H contributed from water in the cultivation medium ($x_W$) and
the net hydrogen isotope fractionation between fatty acids and water ($\alpha_{fa/w}$), where $a_W = x_W \times \alpha_{fa/w}$. Whereas $a_W$ can be
determined experimentally, the latter terms cannot be independently determined for heterotrophic growth (Kopf et al., 2015).



The traditional isotope effects and $\varepsilon_{C18:2/water}$ and $\alpha_{C18:2/water}$ were calculated after Hayes (2004), where $\alpha_{C18:2/water} = [(\delta^2 H_{C18:2} +$
$1000) / (\delta^2 H_{water} + 1000)]$ and $\varepsilon_{C18:2/water} = (\alpha_{C18:2/water} - 1) \times 1000‰$.
**3 Results**
**3.1 Fungal growth and $CO_2$ production**
All fungal species were pure cultures, which were incubated in a mineral medium with either glucose, succinate, β-cyclodextrin,
or tannic acid serving as the sole organic C source. Growth was monitored by the evolution of $CO_2$ into the headspace, which
ranged from 0.36% (no respiration of substrate) to a maximum of 35%, after incubation times ranging from 5 to 160 days (Fig.
1). The pH of the media in all incubations ranged from 2 to 5.5 at the time of harvest, with a general trend of decreasing pH
with increasing $CO_2$; however, the trend was opposite when succinate was the carbon source, with pH increasing from 4 to 5.5.
For samples that produced sufficient biomass, the dry biomass of harvested fungal hyphae ranged up to 250 mg, and at least
30 µg dry biomass was used to analyze fungal membrane fatty acids by Pyr-GC-IRMS. Only the Ascomycota species PL and
PJ grew sufficiently on each tested substrate to produce enough biomass for stable isotope analysis. Incubations of Zygomycota
species with glucose or succinate also yielded sufficient dry biomass, and only UM was able to grow on β-cyclodextrin.
Zygomycota species produced neither $CO_2$ nor biomass when incubated with tannic acid. The Basidiomycota typically
exhibited the slowest growth, and both species (PI and PC) only produced enough biomass when grown on glucose. The
headspace $CO_2$ levels in Basidiomycota incubations with succinate increased to a maximum of ~ 2%, but only PI yielded
sufficient biomass for analysis. PC grew sufficiently on β-cyclodextrin, with $CO_2$ levels increasing to a maximum of 3%, while
$CO_2$ remained < 0.6% in PI incubations.

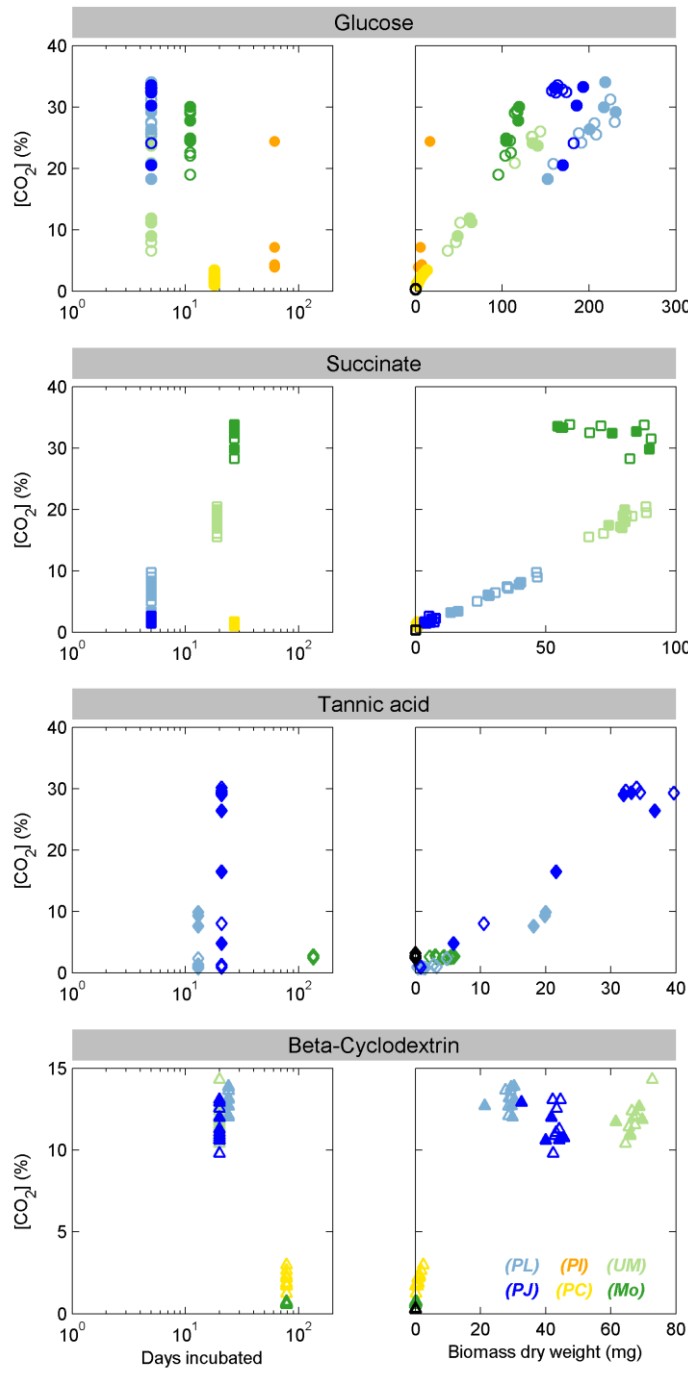

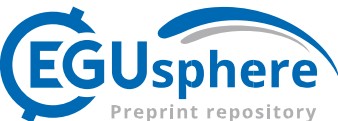

**Figure 1. Growth of fungal species on each substrate as indicated by production of CO₂ versus days of incubation (left panels) or dry**
**biomass (right panels). Filled symbols indicate samples for which the $C_{18:2}$ biomarker was measured by Pyr-GC-IRMS. Colors**
**represent the Ascomycota species *Penicillium janczewskii* (PJ, dark blue) and *Paecilomyces lilacinus* (PL, light blue), Zygomycota**
***Mortierella* sp. (MO, dark green) and *Umbelopsis* sp (UM, light green), and Basidiomycota species *Phanerodontia chrysosporium* (PC,**
**orange) and *Paxillus involutus* (PI, yellow). The symbols denote incubations with glucose (circles), succinate (squares), tannic acid**
**(diamonds), or β-cyclodextrin (triangles).**

The growth substrates induced a wide range in CUE values, ranging from 0.1 to 0.6 (Fig. 2). Higher CUE values were typically
observed for Ascomycota and Zycomycota species growing on glucose, and lower values for their growth on succinate and
tannic acid. CUE estimated for Basidomycota species was always low (< 0.15). The CUE range for growth on glucose (0.1-
0.6), β-cyclodextrin (0.1-0.6), and succinate (0.2-0.5) were larger than observed for tannic acid (0.15-0.3).

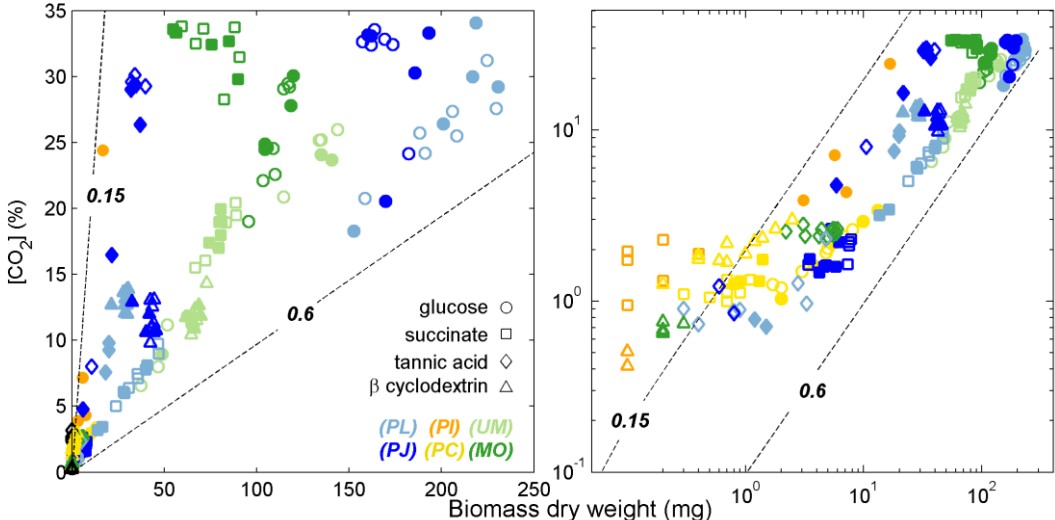


**Figure 2. Accumulation of biomass and headspace CO₂ for fungal species from three phyla growing on monomers or**
**complex substrates. The right panel includes the same data on a log-log scale to depict the trends of fungi exhibiting**
**minimal growth. Lines indicate CUE trajectories of 0.15 or 0.6, and were calculated assuming that fungal biomass**
**was 44% C (w/w). Colors and symbols are redundant with Fig. 1. Filled symbols indicate samples that were analyzed**
**by Pyr-GC-IRMS.**
**3.2 Stable isotopic composition of fungal lipids**
Fungal respiration of the different (unlabeled) growth substrates led to decreasing $\delta^{13}C\text{-}CO_2$ values as fungal biomass was
produced, which followed a hyperbolic trend expected for the mixing of CO₂ from two different sources (Text S1; c.f., Kendall

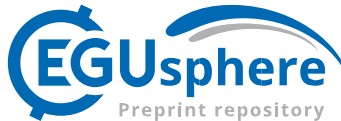

and Caldwell, 1998). The atom % $^{13}C$ in control incubations with no fungal inoculum was measured at the latest time of harvest
of inoculated incubations and stayed below 0.4%, except for incubations with tannic acid, where it ranged between 2% and
3%; the $\delta^{13}C$ values of the substrates were glucose = -26.5‰; succinate = -28.3‰, tannic acid = -27.4‰, β-cyclodextrin = -
10.6‰. The mixing relationship was modeled using all $CO_2$ data, across all incubations, and integrated to approximate the
mixing-weighted average $^{13}F$ value of IC for each incubation (cf., Text S1, Fig. S2), which was finally applied in the
denominator of Eq. 1 to estimate the fraction of lipid-C derived from IC. For incubations that produced sufficient fungal
biomass for stable C isotopic analysis, the weighted average $\delta^{13}C$ values of IC that were applied in Eq. 1 ranged from 200 to
1400 ‰ (i.e., ~ 1.3 to 2.6 AT% $^{13}C$), and was largely dependent on how much of the substrate was respired to $CO_2$.

**3.2.1 Carbon isotopes**
The $\delta^{13}C$ values of fungal biomarker fatty acid $C_{18:2}$ (Table 1) produced under natural cultivation conditions with glucose (i.e.,
non-labeled; AT%$_{DIC}$ ~ 1%) ranged from -24.1‰ to -21.2‰ across all strains (n = 6 species). As expected, $C_{18:2}$ harvested from
the labeled incubations exhibited slightly higher $\delta^{13}C$ values (up to +11‰; PC grown on glucose) than the corresponding
experiment amended with natural bicarbonate, likely owing to the incorporation of labeled IC into the $C_{18:2}$ fatty acid.

**Table 1: $\delta^{13}C$ values of fungal biomarker $C_{18:2}$ harvested from incubations with non-labeled substrates (nat) or those**
**amended with $^{13}C$-labeled bicarbonate. Incorporation of inorganic C (%IC) was calculated based on Eq.1. Errors**
**represent the standard deviation of replicate incubations. Not all fungal species grew sufficiently on all substrates and**
**thus %IC could not have been determined (n.d).**

| Species | Glucose $\delta^{13}C$ (‰) | | IC (%) | Succinate $\delta^{13}C$ (‰) | | IC (%) | Tannic acid $\delta^{13}C$ (‰) | | IC (%) | β-cyclodextrin $\delta^{13}C$ (‰) | | IC (%) |
|---|---|---|---|---|---|---|---|---|---|---|---|---|
| | nat | + | | nat | + | | nat | + | | nat | + | |
| Paxillus involutus (PI) | -21.5 | -16.7 | 0.6 (±0.2) | n.d | n.d | n.d | n.d | n.d | n.d | n.d | n.d | n.d |
| Phanerodontia chrysosporium (PC) | -24.1 | -15.9 | 0.9 (±0.3) | -27.3 | -23.5 | 0.3 (±0.1) | n.d | n.d | n.d | n.d | n.d | n.d |
| Mortierella (MO) | -21.9 | -20.7 | 0.5 (±0.1) | -31.7 | -31.5 | 0.1 (±0.1) | n.d | n.d | n.d | n.d | n.d | n.d |
| Umbelopsis (UM) | -21.2 | -21.1 | 0.1 (±0.0) | -30.6 | -28.2 | 0.7 (±0.2) | n.d | n.d | n.d | -21.7 | -18.1 | 0.8 (±0.3) |
| Penicillium janczewskii (PJ) | -23.2 | -22.1 | 0.5 (±0.2) | -30.1 | -27.8 | 0.2 (±0.0) | -25.9 | -20.5 | 2.2 (±0.5) | -20.7 | -20.1 | 0.1 (±0.2) |
| Paecilomyces lilacinus (PL) | -23.0 | -23.5 | n.d | -30.8 | -29.9 | 0.1 (±0.0) | -25.4 | -25.0 | 0.1 (±0.0) | -19.1 | -17.6 | 0.7 (±0.4) |




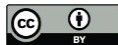

The estimated incorporation of IC into $C_{18:2}$ (%IC) typically ranged up to 1%; only PJ grown on tannic acid exhibited higher
%IC values, which ranged up to 2.2% (Table 1, Fig. 3). There were no general trends observed in %IC with other measured or
estimated parameters, including CUE; however, for the two species that were able to grow on tannic acid, %IC was positively
correlated with the amount of $CO_2$ and biomass produced during the incubation ($R^2 > 0.85$, n = 5, p < 0.01).

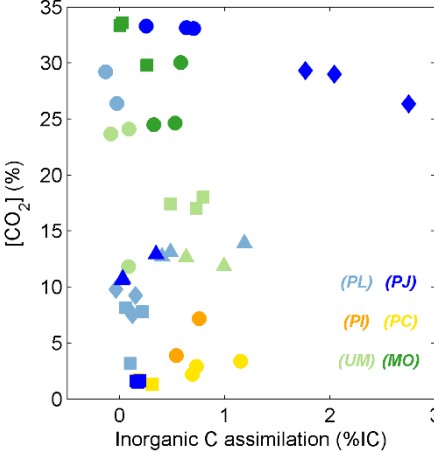

**Figure 3. %IC values for fungal species respiring glucose (circles), succinate (squares), tannic acid (diamonds), or β-**
**cyclodextrin (triangles). Colors represent individual fungal isolates as described in Fig. 1.**
**3.2.2 Water assimilation factor ($a_W$)**
The "net" contribution of water hydrogen to lipid H is reported as the water hydrogen assimilation factor $a_W$ (Kopf et al., 2015),
and was estimated based on the slope of the linear regression line between H isotopic composition of lipid versus growth
medium water (Fig. 4), which ranged from natural MilliQ ($\delta^2H$ = -45‰ ± 10‰) to the labeled treatments (65‰ ± 4‰; 166‰
± 10‰; 368‰ ± 27‰). The $a_W$ values for the fungal biomarker $C_{18:2}$ grown on glucose ranged from 0.37 ± 0.03 to 0.75 ± 0.06
with an average value of 0.60 ± 0.05 (n = 6 species; ±SEM). When grown on succinic acid, the $a_W$ values for $C_{18:2}$ harvested
from individual species ranged from 0.78 ± 0.01 to 0.96 ± 0.02 with an average value of 0.83 ± 0.04 (n = 4 species; ±SEM).
When grown on tannic acid, the $a_W$ values for $C_{18:2}$ harvested from individual species ranged from 0.74 ± 0.06 to 0.77 ± 0.03,
and when grown on β-cyclodextrin the $\alpha_W$ values for $C_{18:2}$ ranged from 0.46 ± 0.03 to 0.68 ± 0.04 with an average value of
0.58 ± 0.06 (n = 4 species; ±SEM). The average $a_W$ values for $C_{18:2}$ for all substrates and species was 0.67 ± 0.04 (±SEM).



The range of traditionally reported isotope effects $\alpha_{C18:2/water}$ and $\varepsilon_{C18:2/water}$ (Sessions and Hayes, 2005) for all natural and $^2$H-
labeled fungal growth experiments was 0.73 to 1.08 and -265 to +83 ‰, respectively. PL growth on tannic acid exhibited the
highest values (0.97 to 1.08 and -35 to +83 ‰, respectively; Fig. 5, Table S1), while all other growth experiments $\alpha_{C18:2/water}$
and $\varepsilon_{C18:2/water}$ remained < 0.94 and -65 ‰, respective. The average (±SD) $\varepsilon_{C18:2/water}$ values were not significantly different for
fungi growing on glucose (-151 ± 121 ‰, n = 6), succinate (-121 ± 44 ‰, n = 4), tannic acid (-39 ± 63 ‰, n = 2), or β-
cyclodextrin (-171 ± 82‰, n = 3).


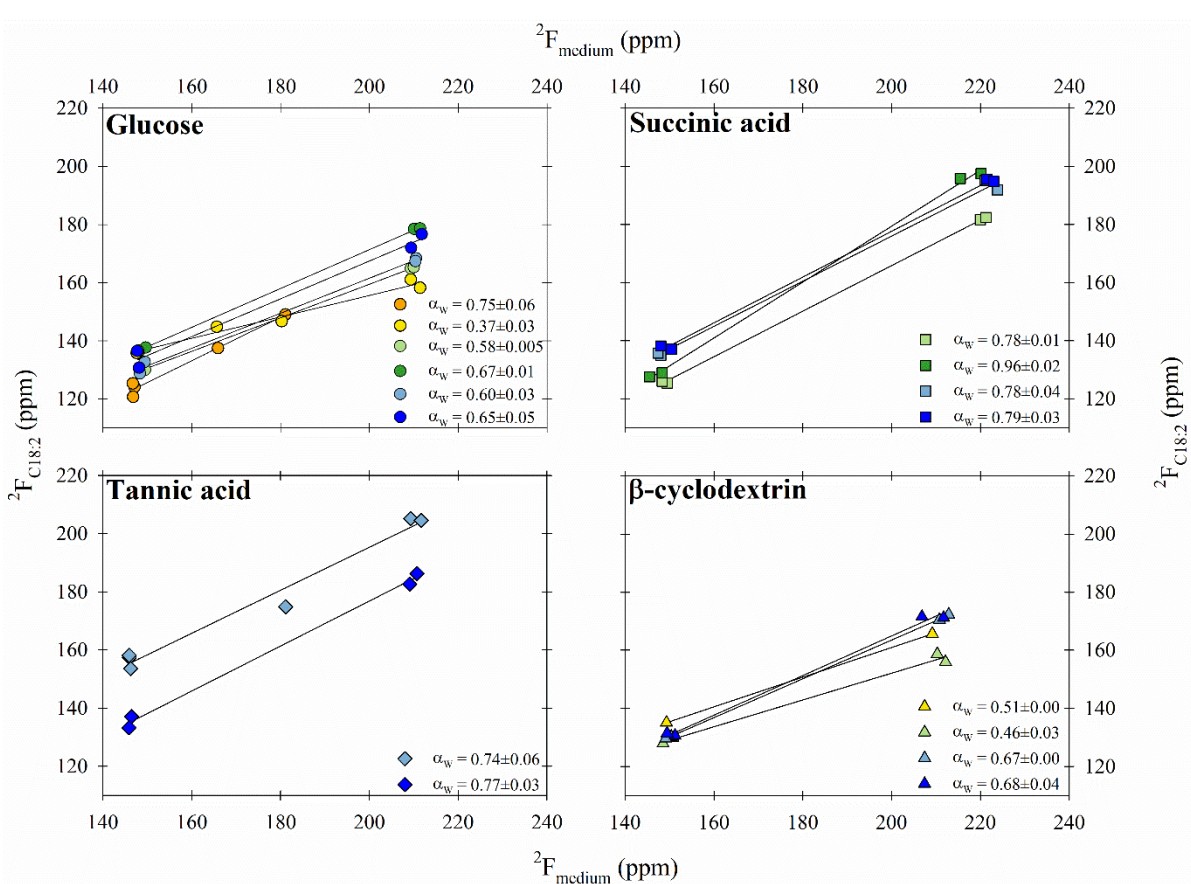


**Figure 4. The water hydrogen assimilation factor ($a_W$ values) estimated as the slope of the fractional $^{2/1}$H abundance ($^2$F) in lipids (y-**
**axis) versus medium water (x-axis). Data are shown for fungal biomarker $C_{18:2}$ produced during growth on the different substrates**
**(glucose, succinic acid, tannic acid and β-cyclodextrin) and harvested from the different fungal isolates [*Paxillus involutus* (PI),**
***Phanerodontia chrysosporium* (PC), *Mortierella* sp. (MO), *Umbelopsis* sp. (UM), *Penicillium janczewskii* (PJ), and *Paecilomyces***
***lilacinus* (PL)]. $R^2$ values for all slopes were > 0.97.**



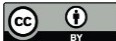


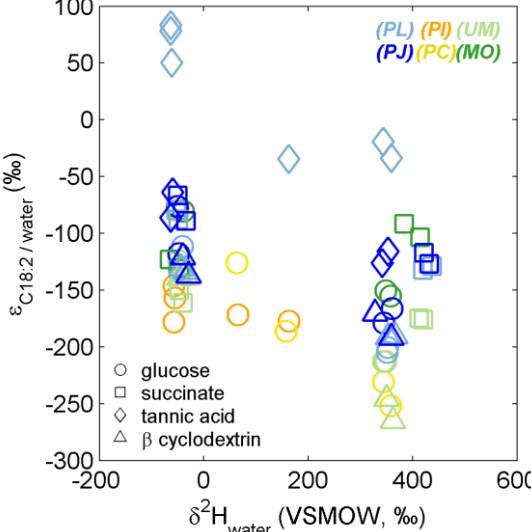


**Figure 5. The apparent isotope effect ($\varepsilon_{C18:2/water}$) for fungi grown in medium having variable $\delta^2H_{water}$ composition. The colors and symbols are redundant with Fig. 1. For each species-substrate pair, the large range and/or decrease in $\varepsilon_{C18:2/water}$ in $^2H$-enriched medium are expected, given the additional contributions of substrate-H and metabolic water-H to lipid-H during biosynthesis.**

## 4 Discussion

### 4.1 Fungal growth dynamics

Collectively, the fungal incubation experiments included six species representing three different phyla growing on one of four substrates, and exhibited a large range in the relative amounts of $CO_2$ (0.2-34% v/v) and biomass produced (0-230 mg dry weight; Fig. 1), with CUE ranging from 0.15 to 0.6 (Fig. 2). While atmospheric, oxic conditions likely prevailed during most of the incubation period, it is probable that some incubations turned anoxic when $CO_2$ levels exceeded 21%, which occurred in incubations of Ascomycota and Zygomycota growing on glucose, Ascomycota growing on tannic acid, and Zygomycota growing on succinate. The accumulation of $CO_2$ was necessary consequence of performing the incubations in closed bottles, which was required to prevent the escape of $^{13}C$-labeled IC and also to prevent microbial contaminations. Nevertheless, such alteration between oxic and anoxic conditions is common in natural environments, and the measured IC assimilation into fungal lipids was consistently low (< 3%; Fig. 3), regardless the implied anoxia. Furthermore, the observed variability in CUE, %IC, and $a_W$ between growth experiments were not correlated with large increases in headspace $CO_2$.

302



**4.2 Fungal IC assimilation into lipids**

A fundamental process in nature and basis for ecological food webs is the fixation of IC via photosynthesis and/or chemosynthesis by autotrophic organisms. The IC assimilation by heterotrophic organisms also plays an important role in ensuring the provision of energy and to replenish intermediates in the TCA cycle that have been released for biosynthesis (Kornberg 1965). Therefore, IC assimilation is a proxy for both anabolic processes and the catabolic status of the cell, influenced by assimilation, biosynthesis, anaplerotic reactions, and redox balancing reactions (Braun et al., 2021; Erb 2011). Previous reports on the by-fixation of IC (%IC) via anaplerotic pathways into heterotrophic biomass varied between 1% and 8% (Dijkhuizen & Harder, 1985; Feisthauer et al., 2008; Romanenko 1964; Roslev et al., 2004), whereas for fungi it was previously reported to amount to roughly 1% (Sorokin 1961; Schinner & Concin, 1981; Schinner et al., 1982), and was recently shown to vary between 2% and 12% for Ascomycota when grown on glucose or glutamic acid (Jabinski et al., 2024). Our results, focusing on a specific fatty acid biomarker, demonstrate a low range in %IC for all different substrates and species tested in this study (0 - 3%), with the Ascomycota (0 - 2%) assimilating relatively less IC than previously reported species (4.6% ± 1.6%; Jabinski et al., 2024). The highest observed incorporation was 2.2 ± 0.5% by *Penicillium janczewskii* (PJ, n = 3) when grown for 21 days on tannic acid (Table 1; Fig. 3). Only the other Ascomycota species, *Paecilomyces lilacinus* (PL), grew sufficiently on tannic acid (up to 10% $CO_2$ and 20 mg dry weight after 13 days; %IC = 0.14 ± 0.02%, n = 3 ; Figs. 1, 3), suggesting that increased assimilation of IC by PJ may have occurred during the extra week of incubation and promoted higher biomass production. The high $CO_2$ levels also suggest that the incubations of PJ with tannic acid may have turned anoxic, which may also explain the higher IC incorporation in these incubations. Overall, heterotrophic IC assimilation does not appear to be a hallmark of any of the variety of fungal taxa or catabolic pathways probed in this study.

**4.3 Water hydrogen incorporation into fungal lipid biomarker C$_{18:2}$**

As demonstrated previously, the regression slope between hydrogen isotopic composition of water medium and microbial lipids (i.e., $a_W$) varies with the type of metabolism (Zhang et al., 2009; Valentine, 2009; Wijker et al., 2019; Jabinski et al., 2024). For fatty acid biosynthesis, H incorporation is suggested to be a function of transporters and electron acceptors (NADPH and NADH), with contributions accounting for around half of all lipid hydrogen (Maloney et al., 2024). The remaining comprises equal contributions of H obtained directly from environmental water or acetyl-CoA (Valentine, 2009; Zhang et al., 2009; Caro et al., 2023). The consensus from previous studies that investigated the lipids of heterotrophic bacteria is that microbial heterotrophs exhibit $a_W$ values ranging from 0 to 1, with a mean of 0.71±0.17 (e.g., Caro et al., 2023), though some organisms have exhibited $a_W$ values exceeding 1 (Dirghangi et al., 2013; Jabinski et al., 2024). Jabinski et al. (2024) demonstrated that five species of heterotrophic Ascomycota exhibit similar $a_W$ values (0.62 ± 0.04) for the fungal biomarker C$_{18:2}$ during growth on glucose. Zhang et al. (2009) reported similar $a_W$ values for *E.coli* grown on glucose (0.63 ± 0.03). In the current study, $a_W$ values for the fungal biomarker C$_{18:2}$ during growth on glucose (0.60 ± 0.05) were agreeable with Jabinski et al. (2024), but



more variable, likely owing to the broader phylogenetic coverage of the current study. The similarly large variability in
$\varepsilon_{C18:2/water}$ values can be partly attributed to the large range in $\delta^2H$ of medium water, which contributes H together with the
substrate and metabolic water to determine $\delta^2H_{C18:2}$, even though the net isotope effect ($\alpha_{C18:2/water}$) may be consistent for a
specific species-substrate pair (Fig. 5; Session and Hayes, 2005; Kopf et al., 2015). In other words, $\delta^2H_{C18:2}$ of heterotrophic
fungi exhibits more inertia than $\delta^2H_{H2O}$ in highly labeled incubations, yielding lower $\varepsilon_{C18:2/water}$ values compared to natural
abundance incubations. More robust differences were observed in $a_W$ values both between and within the different phyla and
substrates tested.

*4.3.1 Trends across fungal phyla*
Ascomycota exhibited the most consistent $a_W$ values among phyla when grown on each of the four different substrates [0.63 ±
0.03 (glucose); 0.78 ± 0.01 (succinate); 0.76 ± 0.02 (tannic acid); 0.67 ± 0.01 (β-cyclodextrin)], but also the largest variability
in CUE (0.08-0.59; Fig. 2). CUE and $a_W$ were not significantly correlated across incubations of Ascomycota, suggesting that
drastic changes in the central metabolic pathways that convert C substrates to either biomass or energy reserves did not
systematically alter the net water-H incorporation into lipids. This in in contrast to the prevailing notion that $a_W$ values respond
to changes in NADPH production and turnover within the cell (e.g. Wijker et al., 2019). Basidiomycota only produced
sufficient biomass when fed mycotasubstrates that activated the glycolytic pathway (glucose or β-cyclodextrin; CUE < 0.3),
yet showed high variability in $a_W$ between species ($0.37 \pm 0.03 < a_W < 0.75 \pm 0.06$; Fig. 4), which were beyond the more
confined range of $a_W$ values determined for isolates belonging to Zygomycota and Ascomycota. For incubations in which the
investigated Zygomycota isolates produced enough biomass to determine $a_W$ (i.e., glucose, succinate, β-cyclodextrin; *n* = 5;
Fig. 4), we observed a highly significant inverse correlation with CUE ($R^2 = 0.87$, p < 0.01; Fig. 6), suggesting the potential
for $a_W$ to serve as a proxy for growth efficiency.

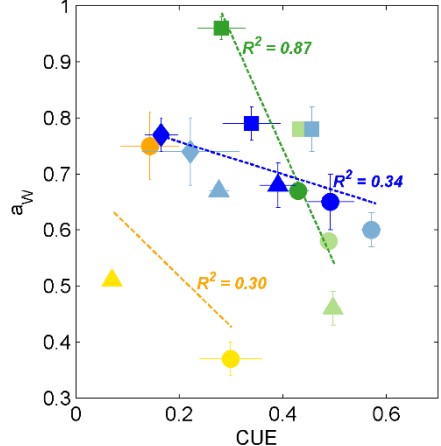




**Figure 6. Biplot of CUE and a$_W$ values for fungi growing on glucose (circles), succinate (squares), tannic acid**
**(diamonds), or β-cyclodextrin (triangles). The colors are redundant with Fig. 1, representing the phlya Ascomycota**
**(blue shades, *n* = 8), Zygomycota (green shades, *n* = 5), and Basidiomycota (yellow and orange symbols, *n* = 3). The**
**dashed lines and $R^2$ values indicate the linear regression for the corresponding phylum across all substrates that yielded**
**sufficient biomass. Only the regression for Zygomycota was significant (p < 0.01).**

*4.3.1 Trends across C substrates*
Across all incubations, the similar a$_W$ values determined for growth on glucose (0.60 ± 0.05) versus β-cyclodextrin (0.58 ±
0.06), of which the latter consists of seven glucopyranose units ($C_6H_{12}O_6$), suggests that the catabolism of glucose subunits via
glycolysis overprints signals of water-H incorporation that may derive during degradation of the β-cyclodextrin oligomer.
Alternatively to glycolysis, succinate yielded significantly higher a$_W$ values (0.83 ± 0.05), which was in the same range as
reported for *E.coli* when grown on succinate (a$_W$ 0.80 ± 0.05; Zhang et al., 2009), and was more similar to that reported
previously for fungal growth on glutamic acid (0.90 ± 0.07; Jabinski et al., 2024). Considering all fungal incubations, a one-
way analysis of variance (ANOVA; Holm-Sidak method; SigmaPlot v11) confirmed the significant difference in a$_W$ values
between growth on glucose and glutamic acid (p < 0.001), glutamic acid and β-cyclodextrin (p < 0.001), succinate and glucose
(p < 0.003), and succinate and β-cyclodextrin (p < 0.005). It also confirmed that there was no significant difference between
the other substrate combinations (p > 0.05).
Glutamic acid and succinate are thought to be introduced into the TCA cycle through coupled metabolites, where succinate is
a direct metabolite of the TCA cycle and glutamic acid is converted to α-ketoglutarate by transamination before entering the
TCA cycle, which is only 2 steps from succinate (Cooper et al., 2014). Also, being acids, these substrates may have a greater
capacity than saccharides to exchange H with ambient water at experimental pH (typically 2 < pH < 5.2), especially glutamic
acid, which also comprises an amino moiety. Tannic acid (0.76 ± 0.02) yielded no significant differences (p > 0.05) from the
other substrates, and is reported to be degraded to different subunits including gallic acid and glucose (Banerjee and Mahapatra,
2012; Lekha and Lonsane, 1997 and references within). Aromatic degradation pathways employed by fungi generate
intermediates that go through the β-ketoadipate pathway (Mäkelä et al., 2015) before entering the TCA cycle as a succinyl-
CoA metabolite (Lekha and Lonsane, 1997). The a$_W$ values induced by degradation of tannic acid suggest that it integrates
both the lower a$_W$ signature of glycolysis and higher a$_W$ signature of the TCA cycle (Fig. 7). Similarly to trends in a$_W$, the
ε$_{C18:2/water}$ values estimated for fungal growth on succinate or tannic acid were typically higher than glucose or β-cyclodextrin,
for a given $\delta^2H_{H2O}$ treatment (Fig. 5; Table S1).
Together, our incubation experiments suggest that a$_W$ values determined for the fungal biomarker $C_{18:2}$ could not distinguish
between fungal growth on relatively labile monomers (i.e., glucose and succinate; requiring as few as 5 days of cultivation)
versus larger, less-labile substrates (i.e., β-cyclodextrin and tannic acid; requiring 20 to 183 days of cultivation). However, a$_W$
values of fungal lipid biomarkers may be indicative of fungi employing primarily glycolytic or TCA pathways. Environmental

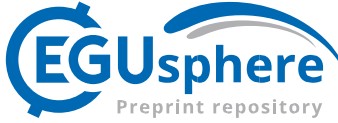

assays that quantify fungal lipid production via the incorporation of ambient water-H (i.e., the lipid-SIP approach) may upscale
to total production estimates by applying our calculated mean $a_W$ value of $0.69 \pm 0.03$ [n = 27; $\pm$ (SEM)], which is consistent
with the $a_W$ value of 0.71 recommended for soil microbial communities (Caro et al., 2023). For $^2$H-SIP investigation of fungal
ecotypes supplied with TCA metabolites, such as mycorrhiza, the $a_W$ value of $C_{18:2}$ may range up to $0.83 \pm 0.05$. Similar
approaches could be applied to environmental samples, such that the $a_W$ values of phospholipids containing C18:2 fatty acids
could inform the distribution of predominant metabolic ecotypes across a soil profile.

### 4.4 Dual-SIP approach

Dual-SIP experiments with $^2$H$_2$O and $^{13}$C-dissolved IC previously highlighted the potential to track microbial activity and
distinguish heterotopic vs autotrophic metabolic modes within environmental settings and pure cultures (Kellerman et al.,
2012, 2016; Wegener et al., 2012; Huguet et al., 2017; Wu et al., 2018, 2020). This approach was also previously applied to
investigate fungal pure cultures (Jabinski et al., 2024), in which the plot of assimilation of IC versus water-H into the fungal
biomarker $C_{18:2}$ could distinguish five Ascomycota species growing on glucose or glutamic acid, with $a_W$ values explaining
most of the variability. While calculated IC:$a_W$ are useful to distinguish autotrophic from heterotrophic growth (cf. Wegener et
al., 2016), all calculated values in this study remained near zero, with %IC ranging up to 3% and $a_W$ values ranging from 0.37
to 0.96 (Fig. 4). This pure culture study therefore suggests that fungal assimilation of IC is low and less insightful than the
more distinguishable $a_W$ values for identifying the relative contributions of fungal phylotypes or ecotypes in environmental
assays.

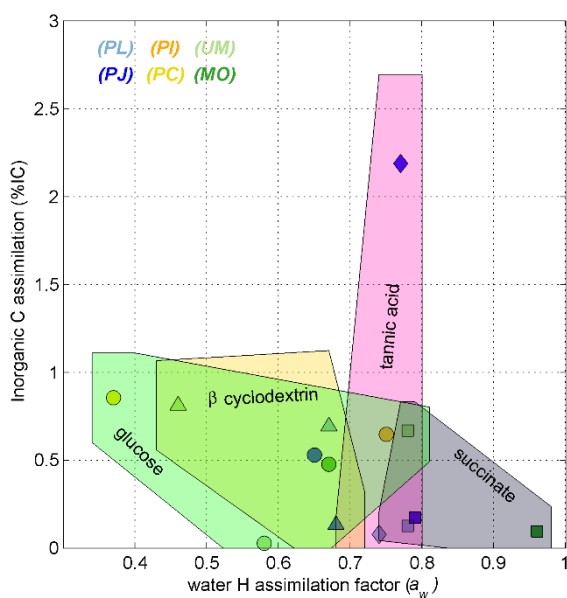




**Figure 7. Scatterplot of %IC and $a_W$ values plot for $C_{18:2}$ harvested from incubations of Ascomycota (blue symbols), Basidiomycota (yellow and orange symbols), and Zygomycota (green symbols). The shaded polygons span the range of %IC and $\alpha_W$ values (± propagated error) for fungal growth on glucose (circles, green shape), succinate (squares, grey shape), tannic acid (diamonds, pink shape), or β-cyclodextrin (triangles, yellow shape). Each data point represents one species-substrate pair determined for $n > 4$ growth experiments.**

## 5 Conclusion

The purpose of this work was to apply the dual-SIP assay on pure fungal cultures to define the effect of different organic C substrates on the incorporation of water-H and IC into their membrane lipids. Although heterotrophic $CO_2$ fixation by microbes may range up to 8% of biomass C, the IC assimilation into the fungal biomarker $C_{18:2}$ harvested from six species representing Ascomycota, Basidiomycota, and Zygomycota did not vary consistently between species or substrate, and remained below 3%. Our findings suggest that the fungal catabolic pathways activated by the variety of substrates tested in this study cannot fully account for the higher levels of heterotrophic $CO_2$ incorporation reported in natural systems. However, *Penicillium janczewskii*, the species that was most successful at respiring tannic acid, also exhibited the highest %IC value of all incubations (Fig. 3; Fig. 7), suggesting that fungal degradation of similarly complex substrates may rely in part on the assimilation of IC (e.g., via anaplerotic reactions). The use of SIP to estimate %IC of heterotrophs required (i) a closed system to prevent loss of [13]C label to the atmosphere, and (ii) a high label dose, to contend with the accumulation of $CO_2$ respired from growth substrate. These conditions intensified upon decreases in pH during the growth experiments, thereby shifting the speciation of IC toward $CO_2$, which may have further, yet unknown consequences for anaplerotic incorporation of the IC. Future applications to determine %IC for microbial heterotrophs should consider repeated spiking or continuous cultivation practices to better stabilize pH and $\delta^{13}C_{DIC}$. Likewise, it would be worth considering isotopic analyses of other potential bioindicator compounds (e.g., sterols, peptides, aminosugars) to see whether incorporation of the IC into such other compounds would render higher levels than fatty acids.

In contrast to %IC, we conclude that substrates that activated the glycolysis pathway yielded significantly lower $a_W$ values than those catabolized as TCA intermediates. The expanded dataset reported in this study suggests that the accuracy of fungal production estimated by [2]H-lipid SIP experiments can be improved by applying the average $a_W$ value of 0.69 for saprotrophic fungi or up to 0.83 for mycorrhizal fungi. Furthermore, determination of $\alpha_W$ values in environmental [2]H-SIP assays may be useful to identify the relative contributions of fungal ecotypes that rely on C substrates fueling glycolysis (e.g., leaf litter) versus those that are fed primarily by TCA intermediates (e.g., root or microbial exudates). Lastly, to our knowledge, the two Zygomycota strains investigated in this study provide the first evidence of a potential correlation between $a_W$ and CUE (Fig.



6), encouraging further exploration of the link between these two parameters, both of which are coupled to microbial central
metabolic pathways.

**Data availability**
Data presented in the figures and tables are available in supplementary material (Table S1) and will be made available on the
Fractome Database (https://fractome.caltech.edu/).

**Author contribution**
Stanislav Jabinski, Conceptualization, Data curation, Formal analysis, Investigation, Methodology, Software, Validation,
Visualization, Writing – original draft, Writing – review and editing
Vítězslav Kučera, Investigation, Methodology, Resources, Writing – review and editing
Marek Kopáček, Formal analysis, Methodology, Resources, Validation,
Jan Jansa, Conceptualization, Formal analysis, Methodology, Resources, Validation, Writing – review and editing
Travis B. Meador, Conceptualization, Data curation, Formal analysis, Funding acquisition, Investigation, Methodology, Project
administration, Resources, Software, Supervision, Validation, Visualization, Writing – original draft, Writing – review and
editing
**Competing interests**
The authors declare that they have no conflict of interest.
**Disclaimer Acknowledgements**
We thank Ljubov Poláková for the support of the stable isotope measurements and laboratory protocols; the Collection of
Microscopic Fungi of the Institute of Soil Biology BC CAS for providing the fungal species *Penicillium janczewskii* strain
BCCO20_0265 and the Institute of Institute of Microbiology CAS for providing the fungal species *Paxillus involutus* strain
SB-22; *Phanerodontia chrysosporium* strain CCM8074, *Mortierella* strain RK-38; *Umbelopsis* strain RK-43 and *Paecilomyces*
*lilacinus* strain DP-23. This Project was funded by the Czech Science Foundation GACR nr. 20-223805 (FUNSIF) and
supported by MEYS CZ grant LM2015075 Projects of Large Infrastructure for Research, Development and Innovations as
well as the European Regional Development Fund-Project: research of key soil-water ecosystem interactions at the SoWa
Research Infrastructure (No. CZ.02.1.01/0.0/0.0/16_013/0001782).


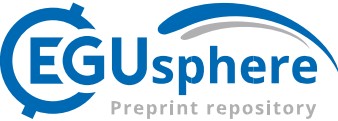






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
