# Peer review of "Fungi present distinguishable isotopic signals in their lipids when"

_EGUsphere, 2024_

## Author Response (AR1)

**Author responses to Reviewer 1 – please find our responses in the blue font below each reviewer comment below.**

In this work, Jabinski and colleagues quantify the water assimilation factors and heterotrophic incorporation of inorganic carbon into biomass, using fungal cultures. This "dual-SIP" approach is useful for calibrating environmental SIP approaches. I have no major comments on this work and recommend it for publication. I have detailed some minor comments below that should be addressed prior to publication.

Thank you for your attention and recommendations to improve the manuscript. We have implemented all suggestions in the revised version of the manuscript.

C1: Section 3.2.2 (and throughout): the water assimilation factor should be noted as a_w, not alpha_w, to avoid confusion with alpha fractionation factor notation.

R1: Thank you for the comment clarifying the text; this is an important issue also raised by the second reviewer. As suggested, we changed the water assimilation factor from alpha to $a_w$ from Section 3.2.2 and throughout the manuscript.

C2: Figure 3: the units of the x and y axes should be cleaned up. For these values I would recommend ppm.

R2: Thank you for the suggestion. We have changed the units to ppm for Figure 3 (Fig. 4 in revised manuscript).

C3: Line 58: Please add the following references here:

Warren 2022 "D2O labelling reveals synthesis of small, water-soluble metabolites in soil."

Caro et al. 2023 "Hydrogen stable isotope probing of lipids demonstrates ..."

Canarini et al. 2024 "Soil fungi remain active and invest in storage compounds during drought independent of future climate conditions"

R3: Thank you for the suggested references, they were added to the corresponding line "a useful tracer of microbial activity in a diverse range of environments (Canarini et al., 2024; Caro et al., 2023; Fischer et al., 2013; Kellermann et al., 2012; Wegener et al., 2016; Warren 2022; Wu et al., 2018)." These references were also added to the revised list of references; Caro et al. was already in the reference list:
Canarini, A., Fuchslueger, L., Schnecker, J., Metze, D., Nelson, D. B., Kahmen, A., ... & Richter, A. (2024). Soil fungi remain active and invest in storage compounds during drought independent of future climate conditions. *Nature Communications*, *15*(1), 10410.
Warren, C. R. (2022). D2O labelling reveals synthesis of small, water-soluble metabolites in soil. *Soil Biology and Biochemistry*, *165*, 108543.

C4: Line 180: Make sure to subscript CO_2.

R4: The error was corrected as suggested.

C5: Line 287: An a_w of 1.2 is nonsensical, no? I was curious and looked at the figure I believe you are referencing, and the histogram bins only go to 1.0. It would be more accurate here to say something along the lines of "A_w can theoretically vary between 0 and 1, representing conditions where an organism acquires none or all of its lipid H from water. The consensus from previous studies is that microbial heterotrophs exhibit a_w values typically around 0.71 +/- 0.17."

R5: Briefly, we agree that it's a very rare occurrence that $a_w$ would exceed value of 1, and have modified the text to cite the typical range. However, there are examples of the literature where $a_w$ was reported to exceed 1 (see below), and we have noted this in the text as well (Line 304).

Regarding theory, although water H assimilation efficiency ($a_w$) can be conceptualized as the proportion of lipid-H derived from water-H (a value ranging from 0 to 1), the term is more complex. In principle, it is not impossible for $a_w$ to exceed 1, as the term integrates several isotope effects during the incorporation of water-H into lipid-H. In any of these steps, the intermediate product may become enriched in the heavy isotope relative to the substrate (inverse isotope effect; e.g., enrichment in 2H of TCA intermediates, [Wijkers et al. 2019; www.pnas.org/cgi/doi/10.1073/pnas.1818372116]; or NADPH residence time, [Torres-Romero et al., 2024, https://doi.org/10.1073/pnas.2318570121]). This signal could be perpetuated into lipid $d^2H$ signals, depending also on the proportion of lipid-H that is derived from NADPH, for example. Post-synthesis modification of fatty acids provides another node for $^2H$ enrichment (e.g., Chikaraishi et al., 2004 (Phytochemistry 65, 2293-2300), where some fatty acids may serve as the substrate in desaturation reactions. An $a_w$ of higher than 1 is rare and has not yet been reported before for bacterial or archaeal organisms, but was already previously reported for eukaryotic organisms such as a ciliate called *T. thermophila*, where the fatty acids as well as another biomolecule demonstrate $a_w$ values exciding 1 [Dirghangi, S. S., & Pagani, M. (2013). Hydrogen isotope fractionation during lipid biosynthesis by Tetrahymena thermophila. *Organic geochemistry*, *64*, 105-111]. A higher $a_w$ value was also reported for fungi [Jabinski, S., d. M. Rangel, W., Kopáček, M., Jílková, V., Jansa, J., & Meador, T. B. (2024). Constraining activity and growth substrate of fungal decomposers via assimilation patterns of inorganic carbon and water into lipid biomarkers. Applied and Environmental Microbiology, 90(4), e02065-23].

C6: Fig. 5: It is unusual to have a figure included in the conclusion section. I would recommend switching the position of this figure to the main text. Furthermore, it is currently unclear what this figure adds to the manuscript. The color and shape scheme is very difficult to parse and should be cleaned up, or the figure should be removed.

R6: The figure has been moved to a new Discussion section in the revised manuscript, section 4.4 Dual-SIP. The shape extends beyond the points displayed as it incorporates the error of the measurements; this information has been added to the figure caption. Text has been added to the figure to clarify the shapes corresponding to each substrate. The color scheme was chosen based on journal requirements for color vision difficiencies, and applies distiguishablecolors feature of Matlab.

**Author responses to Reviewer 2 - please find our responses in the blue font below each reviewer comment below.**

Jabinski et al. cultured fungi from several species with 2H-labeled water and 13C-labeled DIC. The fungi took up very little inorganic carbon, indicating that they were primarily heterotrophs and ate the unlabeled organic carbon provided in the cultures. Apparent $^2H/^1H$ fractionation factors varied among species and substrates, but I don't think the alpha values reported for these were calculated correctly (although what the authors have done is consistent with some other publications, I don't think it is appropriate when the product and the substrate are separated by multiple $^2H$-fractionating reactions, as explained below). The discussion was also a bit shallow in terms of integrating these new results to other publications that have investigated the relationship between metabolism and $^2H/^1H$ fractionation. The emphasis is more on the utility of the dual-SIP approach, and it seems like there is a bit of a missed opportunity to discuss how these results can inform our understanding of fungal metabolism. I think the data are unique and I appreciate that the authors have put in a large amount of work to generate them. Very little is known about 2H/1H fractionation by fungi, and this has the potential to be a useful tool to understand fungal metabolism in soils. I think with some rewriting, this manuscript will be a helpful contribution to this field.

Best wishes,

Nemiah Ladd

We thank the reviewer for her attention and comments to improve our manuscript. We have addressed the discrepancy between the terms $a_W$ and the terms $\alpha_{L/W}$ and $eps_{L/W}$ that are traditionally used to describe kinetic and equilibrium isotope effects, all of which are reported in the revised manuscript and a figure was added to display corresponding eps values (Fig. 5). We expected the fungi cultured for this study to grow heterotrophically and hypothesized that heterotrophic inorganic C uptake (e.g., via anaplerotic reactions rather than explicit autotrophy) would vary depending on the organic C substrate provided for growth; this was not observed however. The revised manuscript further clarifies this hypothesis and finding and provides recommendations for future applications. Additional text and figures take the opportunity to further explore the relationship between $^{2/1}H$ fractionation and fungal metabolism and CUE, in the context of what is known for bacterial heterotrophs.

Specific comments:

C1: Line 27: this fractionation factor is actually much lower than the values typically seen for bacterial heterotrophs, and I think is due to the way you have calculated alpha (see comment below)

R1: Thanks for the comment and highlighting this confusion. The water assimilation factor $a_W$ is not the alpha normally reported for isotope effects, but the net water-H incorporation efficiency, which includes a mass balance of water-H versus other H sources as well as all the different isotope effects happening between the water-H incorporation into fatty acids during lipid biosynthesis. It can be considered as $a_W = x_W * \alpha_{fa/w}$; with $x_W$ being the mole fraction of water derived hydrogen and $\alpha_{fa/w}$ being the traditional isotope effect describing net hydrogen isotope fraction during lipid biosynthesis (Kopf et al., 2015). This equation now appears in the Methods section of the revised manuscript. In the current study, $a_W$ was determined according to Kopf et al., 2015, by culturing fungi in media having four different water isotopic compositions. This allowed us to determine $a_W$ values for the biomarker from the linear regression of $^2F_{biomarker}$ versus $^2F_{water}$. As mentioned in the Methods Section

"The 'net' contribution of water hydrogen to lipid H is reported as the water hydrogen assimilation factor $a_W$ (Kopf et al., 2015), and was estimated based on the slope of the linear regression line between H isotopic composition of lipid versus growth medium water (Fig. 3)". We discuss this topic again in a comment below.

C2: Lines 107, 117, 128, 139 "analytical error": specify if you mean accuracy (measured offset from known value) or "precision" (standard deviation or standard error of replicate standard measurements)

R2: Thank you for the comment, we mean the precision of our measurements and we have changed the term analytical error to precision.

Line 107 "The analytical precision was below 1‰."

Line 117 "Analytical precision of $\delta^2H$ was <1.5‰."

Line 128 "The analytical precision was around 1‰."

Line 139 "The analytical precision was <0.04‰."

C3: Line 166: would be good to specify both precision and accuracy for these measurements. Were samples measured more than once (in duplicate or triplicate)? If so, can you report average standard deviation of replicate sample measurements?

R3: The error reported here is the precision calculated from replicate measurements of the international reference standards USGS 70 and USGS 72 using the same sample introduction technique. Each fungal biomass sample was measured once, but replication was achieved in the form of replicate cultivation experiments (at least two natural and two with $^2H$-enriched medium). With regard to accuracy, we have normalized the data according to the responses of USGS international standards; we note however that even if there is an offset in accuracy, this should apply to all samples such that the slope ($a_W$ value) will not be affected by consistent inaccuracies.

C4: Section 3.2.2, Figure 3: I've seen several instances where $^2H/^1H$ fractionation factors between lipids and water are calculated from the slope in this way, but I don't think it works to calculate alpha from the slope if there is more than one reaction separating the product from the substrate, as is the case for fatty acids synthesized from water. If there was a single fractionating step, the alpha value calculated from the slope would correspond to the epsilon value calculated from the y-intercept. This doesn't seem to be the case for your data. See Sessions, A.L., Hayes, J.M., 2005. Calculation of hydrogen isotopic fractionations in biogeochemical systems. Geochim. Cosmochim. Acta 69, 593–597. It would be better to calculate alpha from each lipid-water pair and report average apparent fractionation factors per species. At a minimum, you should provide the full equations for each linear regression that is shown, including the y-intercept and not just the slope. There is a statement that the data set will be published, but it is not available for reviewers to see, so I'm not able to check this myself.

R4: We believe that there is confusion among the terms $a_W$ defined by Kopf et al. (2015) and the isotope effect described by alpha, which is an alternative notation of the epsilon value [eps = (1-alpha)*1000]; this was also highlighted by Reviewer 1. Over the last decades, researchers have reported the isotope effect between water-H and lipid-H ($eps_{L/W}$) as the combined isotope effects of

the many enzymes and equilibration reactions that ultimately determine $\delta^2$H-lipid during biosynthesis and subsequent modifications (c.f. Sachse et al., 2012), with the understanding that there is no single fractionating step for the incorporation of water H into lipids during biosynthesis; these are now reported in the revised manucript. The term $a_W$ is similar, but represents the product of the traditional alpha and the proportion of lipid-H that is derived from environmental water (Kopf et al. 2015). The latter is not described by the traditional isotope effect alone, at least in theory, as presented in the equations in Sessions and Hayes (2005). $a_W$ is not inherently connected to the $eps_{L/W}$ value via a reversal of the equation noted above, nor does the y-intercept of the regressions of $^{2/1}$H$_{water}$ vs $^{2/1}$H$_{lipid}$ represent the $eps_{L/W}$ value; it would also be influenced by $^{2/1}$H of other H substrates, for example. We have added the equations described by Kopf et al. (2015) and Hayes (2004) to distinguish these terms in the revised Methods section (2.3).

The revised manuscript includes the full regression equation in the figure. The delta values and regression data are now available as a SM table. We have added a figure displaying the corresponding eps values (Fig. 5).

C5: Section 4.1, Figure 4: shouldn't the CUE results be presented already in the results section, not in the discussion as they are here?

R5: The figure and the CUE results are condensed from the results presented in the results section (Fig. 1) and were introduced in the discussion section as supportive evidence. The revised manuscript now presents the CUE values and this figure (now Fig. 2) in the Results section, and further explores these data in the discussion section.

C6: Line 282: see also a recent paper about H isotope fractionation by yeast by Ashley Maloney et al. (PNAS, 2024), which is relevant here since it is one of the few studies to look at H isotope fractionation in fungi

R6: Thank you for bringing this paper to our attention. We included the reference "… H incorporation is suggested to be a function of transporters and electron acceptors (NADPH and NADH), with contributions accounting for around half of all lipid hydrogen (Maloney et al., 2024)."
It has also been added the reference to the reference list:
"Maloney, A. E., Kopf, S. H., Zhang, Z., McFarlin, J., Nelson, D. B., Masterson, A. L., & Zhang, X. (2024). Large enrichments in fatty acid 2H/1H ratios distinguish respiration from aerobic fermentation in yeast Saccharomyces cerevisiae. *Proceedings of the National Academy of Sciences*, *121*(20), e2310771121."

C7: Figure 1, subsequent figures: It would be nice if all six species were indicated in the legend, rather than the three genera. It is a little confusing how half of the colors in the figure are not shown in the legend, and you have to read the caption to make sense of them. Then for the other figures that use the same color scheme, you always have to refer back to the figure 1 caption to figure out what they mean.

R7: We have added the abbreviation for each species in corresponding color to Fig 1,2, 4, and 5. Fig. 3 already contains a lot of text in the figure panel, where $a_W$ values are reported; therefore, the species description appears only in the figure caption.

Technical corrections:

There are several places where articles (e.g., the) are missing or where they are used when they should not be. I've noted some examples of this below, but I suggest that someone read through the manuscript carefully and correct this throughout.

C8: There are also many cases were subscripts or superscripts are missing (e.g., $CO_2$, $^{13}C$)

R8: subscripts and superscripts were checked throughout the manuscript.

C9: Line 39: Add "the" before "atmosphere"

R9: The text has been revised as suggested by the reviewer.

C10: Line 53: Awkward phrasing, I suggest editing to "now allow microbial taxa to be linked to specific processes..."

R10: The text has been revised as suggested by the reviewer.

C11: Line 10: missing articles (need "a" before "standard" and "laboratory"

R11: The text has been revised as suggested by the reviewer.

C12: Line 108: Here, I think you don't need "the"

R12: The text has been revised as suggested by the reviewer.

C13: Line 151: Add "The" before "transfer"; ion source should not be capitalized

R13: The text has been revised as suggested by the reviewer.

C14: Lines 201-208: I would move the equations and the descriptions of them to the methods

R 14: The Lines 201 – 208 were moved into the method section (Lines ~167 – 177)

C15: Line 276: I think you mean to refer to figure 4 here?

R15: Thank you for the correction; yes, the former figure 4 is referred to here. This is now Figure 2 in the revised manuscript.

---

## Author Response (AR2)

**Authors' Response to Reviewers:**

Suggestions for revision or reasons for rejection

(visible to the public if the article is accepted and published)

Jabinski and co-authors have thoroughly revised their manuscript to accommodate the suggestions Reviewer 1 and I made on the previous version. I think the clarity of the text and figures is significantly improved, and especially appreciate the change in terminology from alpha-w to aw, and the additional context provided for how this was calculated – I obviously misunderstood this when I read the original manuscript, which led to some confusion. I realize now that the intention of the study was not really to investigate H isotope fractionation, however I still think that there are some nice results in this regard and that these could be highlighted more clearly. At the moment, the revised version includes some changes that were made in response to my previous comments that get at this a bit, but don't really integrate the H isotope fractionation results with the rest of the manuscript. In some places it is still confusing whether water H assimilation efficiency or H isotope fractionation is being discussed. I guess I also still don't really see why water H assimilation efficiency is so important to focus on in and of itself – the Zhang et al 2009 paper assessed this to provide context for studying H isotope fractionation and the Kopf et al 2015 paper did it in the context of using 2H as a tracer for microbial growth rates. In both cases it is a background calculation buried in the supplement, but here it becomes the focus of the H isotope discussion. The introduction doesn't set up clearly for me why this is important or interesting. However, substrate induced changes to metabolism clearly affect net H isotope fractionation in bacteria (Wijker et al., 2019) and it seems like this study shows similar patterns in fungi. I think this is an interesting finding that could be highlighted more clearly, and better linked to the existing literature on H isotope fractionation – the important papers are cited, but the links and the context are not really clear. I think that with an additional round of minor revisions to the text, this paper will be a very useful contribution and I look forward to seeing it published.

Best wishes,

Nemiah Ladd

We thank Prof. Ladd for improvig this manuscript. She has identified a critical distinction in stable isotope investigations of H isotope fractionation (i.e. $\varepsilon$), which primarily applies to natural systems and paleo records, and water H assimilation efficiency ($a_W$), which mostly applies to stable isotope probing studies. These parallel concepts are now directly addressed in revised Introduction, where we highlight the significance of $a_W$ for SIP applications. These points were also addressed in a recent review [Pilecky, M. et al. 2025, TrAC Trends in Analytical Chemistry, 118194. doi: 10.1016/j.trac.2025.118194], which is cited in the revised manuscript.

The $a_W$ concept was introduced by Zhang et al (2009, PNAS), who presented two figures in the main text to describe the trends, and later empirically determined by Kopf et al. (2015, PNAS), who applied it to estimate microbial growth rates. The importance of the water H assimilation efficiency factor, as questioned above, is underscored by how it contrasts with $\varepsilon_{18:2/water}$ values. As demonstrated in this study (cf. Table 1) and by Zhang et al. (2009), a single species growing on the same substrate under the same conditions can exhibit a broad range of $\varepsilon_{18:2/water}$ values (Figure 1 in this reply), owing, in part, to $^{2/1}H$ kinetic and equilibrium isotope fractionation. Importantly, the $\delta^2H$ value of a lipid or any biomolecule is also determined by the mass balance

of the H that derives from water versus other sources of H; this is especially true for heterotrophs, who incorporate a non-negligible portion of H from their carbon source into lipids. As shown in the figure below, $\varepsilon_{18:2/water}$ estimates converge on two different values, depending on whether the incubation was performed under natural or $^2H_2O$-labeled conditions; hence, neither is an accurate approximation of the true isotope fractionation factor $\varepsilon$, which is independent of $\delta^2H_{H2O}$ in both open and closed systems (Hayes 2004). In contrast to $\varepsilon_{18:2/water}$ values, the water H assimilation efficiency accounts for both $^{2/1}H$ fractionation AND the proportion of H deriving from water, and can therefore be interpreted more robustly as a constant for each species-substrate combination (see Methods). While $\varepsilon_{18:2/water}$ values may remain relevant for natural abundance applications for which there is relatively small variation in $\delta^2H$ of ambient water, the upshot of $a_W$ is that it can be applied to (1) upscale rates of water-H assimilation into fatty acids in SIP studies to achieve a more accurate estimate of total microbial production (e.g. Kopf et al., 2015; Wegener et al., 2012), and (2) assess the metabolic mode of the organism that has produced the biomarker fatty acid, as shown for methanogens, for example (Wu et al., 2020). Similarly to methanogens, fungi are able to metabolize a wide range of substrates. The motivation for this study was to determine $a_W$ values and inorganic C incorporation values for heterotrophic fungi, which will help to inform applications (1) and (2) described above. The corresponding $\varepsilon_{18:2/water}$ values contribute to the mounting database of $^{2/1}H$ fractionation factors described for organisms across all domains of life. While this was not the primary aim of this study, we agree with Prof. Ladd that this estimate is informative and have added text to discuss these findings. We note however that, whereas we performed at least n = 4 independent incubation experiments to determine $a_W$ for each species-substrate combination, n=2 for estimates of $\varepsilon_{18:2/water}$ under natural $^{2/1}H$ abundance conditions.

Please find additional responses to all comments below; the listed line numbers refer to the resubmitted manuscript including track changes.

[Figure]

**Fig. 1.** Variability in $\varepsilon$ values calculated between $^{2/1}H$ compositions of fatty acids and environmental water ($^2\varepsilon_{FA:H2O}$) are inherently coupled to $\delta^2H_{H2O}$. Shown are data for $C_{18:2}$ fatty acid of *Mortierella* grown on glucose (this study; Table S1) and $C_{16:1}$ fatty acid of *C. oxalaticus* grown on acetate (Zhang et al., 2009). Increases in $^2\varepsilon$ values with $\delta^2H_{H2O}$ are a consequence of the changing end-member value (i.e., $\delta^2H_{H2O}$) in the isotopic mass balance, because water-H is not the only source of H to fatty acids. In contrast to $^2\varepsilon$, the water H assimilation efficiency factor ($a_W$) accounts for not only kinetic and equilibrium isotope fractionation between water and fatty acid, but also the relative proportion of H derived from water, and is thus constant for the plotted growth experiments ($a_W$ = 0.67±0.01 and 0.52±0.04, respectively).

Specific suggestions:

Q1: Lines 60-62: This sentence doesn't fit with the flow of this paragraph or make it clear how this is relevant. If I understand correctly, you are mainly using the 2H labeled water to look at how much of the H in lipids comes from water, and not really discussing the differences in 2H/1H

fractionation? But we know that that there are big differences in 2H/1H fractionation with NADPH metabolism, which makes it hard to isolate these two processes. Is that what you are aiming to do here?

R1: This section of the Introduction has been revised to provide more clarity (Lines 60-71). Our aim was to define $^2H$ incorporation into lipids from water, which involves multiple factors. These include, but are not limited to the net $^{2/1}H$ fractionation ($\varepsilon_{lipid/water}$; e.g., via NADPH metabolism, biosynthesis enzymes, etc.) AND the proportion of H ($f_H$) that is derived from water versus alternative H sources (e.g., substrate). These individual terms ($\varepsilon_{lipid/water}$ and $f_H$) are difficult to disentangle, and to our knowledge, cannot be empirically determined for heterotrophs. However, the combined effects of $\varepsilon_{lipid/water}$ and $f_H$ (a.k.a., $a_W$) be identified by the SIP approach applied in this study as described by the equations presented in Kopf et al. (2015). Similarly to H, we aimed to define trends in the inorganic C incorporation for each substrate and species.

Q2: Lines 155-156: The TMSH achieves this by methylating the fatty acids, right? Maybe good to specify this since you switch to describing FAMEs in the next sentence and this way it would be clear where they come from. Also, how do you correct for the isotopic effect of the added C/H from the methyl group?

R2: Thank you for the suggestions to clarify this switch between fatty acids and fatty acid methyl esters; we have revised the sentences accordingly. There was no correction performed for the added methyl group, as the primary objective of this study was the determination of $a_W$ and inorganic C incorporation, for which we compare relative changes in $^{2/1}H$ and $^{13/12}C$, such that knowledge of the stable H and C isotope composition of the additional methyl group is irrelevant (as it is included in FAMEs from both labeled and natural treatments). We have added text to clarify that this correction would improve the accuracy of the reported ε values; however, the high variability ε values for individual growth experiments (~100 ‰; Fig 5, main text) likely far exceeds any shifts in $\delta^2H_{18:2}$ values that may result from the correction.

Line 164-165: "...trimethylsulfonium hydroxide (TMSH) was added on the sample to increase the volatization by transforming the fatty acids into fatty acid methyl esters (FAMEs) and improve measurement sensitivity."

Line 199-201: "Notably, calculation of %IC and aW consider the relative increases in 13/12C or 2/1H composition, and thus do not require a correction for the methyl group added during derivatization of the fatty acid. The εC18:2/water values reported in this study were also not corrected for the additional methyl group and add to the uncertainty of the reported values, as further discussed below."

Q3: Lines 158-159: Did you analyze two separate aliquots of sample biomass (once for H, once for C)? Not clear as written

R3: Yes, this is now clarified in the revised manuscript (Line 167). The samples were analyzed separately for carbon and hydrogen isotopic composition of the same biomass. The conversion to $CO_2$ or $H_2$ for IRMS is stated in the subsequent lines.

Q4: Line 182: Kopf et al based their calculations based on approach used by Zhang et al., 2009, so it would make sense to cite that paper here as well (already cited several other places in the manuscript).

R4: The citation has been added.

Q5: Lines 267-272: This is a different set of results than the water assimilation efficiency, and I think it would make sense to report it with a different sub-heading. I realize this wasn't in the originally manuscript or something that you were focused on before my previous comments. I do think there are some cool results here, showing that the same patterns observed in bacteria are found in fungi, and would be consistent with NADPH metabolism affecting H isotope fractionation during fatty acid synthesis by fungi

R5: A new sub-heading has been added to the revised manuscript.

Q6: Lines 270: It seems like there would be significant differences in epsilon C18:2/water if you compared values for different substrates within single species. There is a lot of differences in fractionation among species, so this gets lost when you look at the average fractionation for each substrate for all species. Within a species, we would expect there to be higher epsilon values for cultures grown on succinate than on glucose for example, which does seem to be the case in your data (it is a little hard to tell from the overlapping symbols in figure 5)

R6: All ε values were reported in Table S1. We now provide statistical comparison (homoscedastic t-tests) only for the non-labeled growth experiments (Lines 287-296), which excludes variability introduced from labeling experiments (discussed in our response above) and is more similar to traditional H isotope fractionation approaches. The significant differences observed between substrates or within species are now reported in the new sub-heading of the Results section; we interpret these with caution given that the number of replicates for each natural incubation are low (n = 2 or 3 for each species-substrate pair).

Q7: Lines 325-342: I still find this paragraph confusing as it is not clear how the discussion of the water hydrogen incorporation relates to the discussion of the fractionation factors
R7: Please refer to our response above. The important point is that NADPH may contribute only "around half of all lipid hydrogen". We have added the following text to the first sentence to provide further clarification:
Line 351: "...and importantly for heterotrophs, a non-negligible proportion of hydrogen may derive from sources other than ambient water."

Q8: Lines 349-350: I think this is a misrepresentation of the Wijker et al paper, which is focused on how changes in NADPH production and turnover within cells affect H isotope fractionation, not net water-H incorporation into lipids. Wijker et al only assessed water incorporation into lipids for growth on glucose (figure S2 of their paper) and found similar rates among the three species of bacteria they tested. There is no discussion of NADPH related to the water assimilation rates that I could find, but quite a lot about how NADPH metabolism impacts net 2H/1H fractionation factors

R8: We have removed this sentence and citation. The revised text now describes the coupling of CUE and $a_W$ as they both relate to the proportion of substrate C and H (i.e., mass balance) that is incorporated into biomass (see Section 4.3.1).

Q9: Line 370: Zhang et al did not grow E. coli on succinate in this study (not listed among the cultures in their Table 1, no data shown for this species/substrate pair)

R9: Thank you for this correction. *C. oxalaticus* was not grown on succinate, it was and it was corrected.

Line 370: "… reported for *C. oxalaticus* when grown on succinate…"

Q10: Conclusions don't say anything about H isotope fractionation during lipid synthesis, even though you have discussed this some throughout the manuscript. I realize this wasn't an original focus of the manuscript, but I think you could make this thread clearer through the intro/results/discussion/conclusion and have some very useful results in this regard.

R10: We have added text to Discussion (Lines 429-439) and Conclusion sections (Line 478-479) to interpret the $\varepsilon_{18:2/H2O}$ values, which are now reported in the Results of the revised manuscript as suggested by this reviewer (see comment above).

Minor comments and typos:

Q11: Line 86: I would use 2H instead of D to be consistent with the rest of the manuscript

R11: Line 86: … and deuterated water ($^2H_2O$).

Q12: Line 121: delete "a" before "12 mL"

R12: …platinum catalyst to 12 mL exetainer vials…

Q13: Figure 4: The alphas in the figure legends should be changed to "a"

R13: The legends in the figure 4 were changed to "a"

References Cited

Hayes, J. M. An introduction to isotopic calculations. Woods Hole Oceanographic Institution, Woods Hole, MA, 2543, 2004.

Kopf, S. H., Sessions, A. L., Cowley, E. S., Reyes, C., Van Sambeek, L., Hu, Y., … & Newman, D. K. Trace incorporation of heavy water reveals slow and heterogeneous pathogen growth rates in cystic fibrosis sputum. *Proceedings of the National Academy of Sciences*, *113*(2), E110-E116, doi: 10.1073/pnas.1512057112, 2016.

Wegener, G., Bausch, M., Holler, T., Thang, N. M., Prieto Mollar, X., Kellermann, M. Y., . . . Boetius, A. Assessing sub-seafloor microbial activity by combined stable isotope probing with deuterated water and 13C-bicarbonate. Environmental Microbiology, 14(6), 1517-1527. doi:10.1111/j.1462-2920.2012.02739.x, 2012.

Wu, W., Meador, T. B., Könneke, M., Elvert, M., Wegener, G., & Hinrichs, K. U. Substrate-dependent incorporation of carbon and hydrogen for lipid biosynthesis by Methanosarcina barkeri. Environmental Microbiology Reports, 12(5), 555567. doi:10.1111/1758-2229.12876, 2020.

Zhang, X., Gillespie, A. L., & Sessions, A. L. Large D/H variations in bacterial lipids reflect central metabolic pathways. Proceedings of the National Academy of Sciences, 106(31), 12580-12586. doi:10.1073/pnas.0903030106, 2009.